# Genotype–Phenotype Association in *ABCC2* Exon 18 Missense Mutation Leading to Dubin–Johnson Syndrome: A Case Report

**DOI:** 10.3390/ijms232416168

**Published:** 2022-12-18

**Authors:** Ji-Hoon Kim, Min-Woo Kang, Sangmi Kim, Ji Won Han, Jeong Won Jang, Jong Young Choi, Seung Kew Yoon, Pil Soo Sung

**Affiliations:** 1The Catholic University Liver Research Center, College of Medicine, The Catholic University of Korea, Seoul 06591, Republic of Korea; 2Division of Gastroenterology and Hepatology, Department of Internal Medicine, College of Medicine, Seoul St. Mary’s Hospital, The Catholic University of Korea, Seoul 06591, Republic of Korea

**Keywords:** *ABCC2*, Dubin–Johnson syndrome, MRP2, hyperbilirubinemia

## Abstract

We report a case of a patient with Dubin–Johnson syndrome confirmed by a genetic study. A 50-year-old woman who had symptoms of intermittent right upper quadrant abdominal pain was diagnosed with calculous cholecystitis at another institute and was presented to our hospital for a cholecystectomy. She had no history of liver disease, and her physical examination was normal. Abdominal computed tomography showed a gallbladder stone with chronic cholecystitis. During a laparoscopic cholecystectomy for cholecystitis, a smooth, black-colored liver was noted, and a liver biopsy was performed. The biopsy specimen showed coarse, dark brown granules in centrilobular hepatocytes via hematoxylin and eosin staining. We performed a genetic study using the blood samples of the patient. In the *adenosine triphosphate-binding cassette subfamily C member 2* (*ABCC2*) mutation study, a missense mutation in exon 18 was noted. Based on the black-colored liver without nodularity, conjugated hyperbilirubinemia, the liver biopsy results of the coarse pigment in centrilobular hepatocytes, and the *ABCC2* mutation, Dubin–Johnson syndrome was diagnosed. The patient was managed with conservative care using hepatotonics. One month after follow-up, total bilirubin and direct bilirubin remained in a similar range. Another follow-up was planned a month later, and the patient maintained her use of hepatotonics.

## 1. Introduction

Dubin–Johnson syndrome (DJS) was first identified by Dubin and Johnson in 1954, introducing a new clinical and pathological diagnosis of chronic jaundice with unidentified pigment in the liver [1]. This condition is characterized by consistent non-hemolytic hyperbilirubinemia with lipochrome-like pigment in liver cells with no signs of hepatic injury [2,3]. Furthermore, it is caused by a mutation in the *adenosine triphosphate-binding cassette subfamily C member 2 (ABCC2)* gene that leads to abnormal excretion of bilirubin from hepatocytes. DJS is harmless, has no long-term complications, and does not need medical attention [2]; thus, such a diagnosis would be necessary for differentiating it from harmful hepatobiliary conditions that may lead to hepatic injury.

DJS is rare and occurs equally among both sexes and all races. The grossly unique “black liver” is characteristic of this disorder, which consists of the accumulation of dark and coarsely granular pigment in centrilobular hepatocytes of a liver with normal parenchymal architecture. Most patients with DJS are asymptomatic, and hyperbilirubinemia is discovered incidentally while undergoing routine health check-ups or being tested for other unrelated health disorders. Rarely, patients may present icterus, general weakness, or abdominal pain.

In this report, we present a case of DJS in a patient who underwent a laparoscopic cholecystectomy due to chronic cholecystitis and was diagnosed via the gross appearance of a black liver, phenotyping (via immunohistochemical and H&E staining), and genotyping (via the detection of a missense mutation).

## 2. Case Report

A 57-year-old woman visited Seoul St. Mary’s Hospital for intermittent right upper quadrant (RUQ) pain. She had visited another hospital for right flank pain and had been diagnosed and treated for a right ureter stone. The computed tomography (CT) scan from the other hospital showed that she also had chronic calculous cholecystitis (Figure 1A). At her visit, she had no gross abnormality on her abdomen and no ongoing symptoms, including pain, dyspepsia, or flank pain. Murphy signs, abdomen tenderness, and rebound tenderness were checked for, but none was found to be positive. Apart from slight elevations in total bilirubin and direct bilirubin, no abnormalities were observed. However, considering the intermittent RUQ pain and chronic calculous cholecystitis on her CT scan, elective laparoscopic cholecystectomy was planned.

The following were the results of the initial liver function tests: aspartate aminotransferase (AST), 13 U/L; alanine transaminase (ALT), 10 U/L; total bilirubin, 1.79 mg/dL; and direct bilirubin, 1.41 mg/dL. Serological tests for hepatitis B and C and autoimmune markers were negative. Moreover, contrast-enhanced abdominal CT revealed no significant abnormality in the liver; only chronic calculous cholecystitis and a few renal cysts could be observed. During her elective laparoscopic cholecystectomy, “black liver” with no nodularity was observed and a liver biopsy was performed by the surgeon (Figure 1B). The laparoscopic cholecystectomy was successfully performed, and no complication from the elective surgery was observed.

The biopsy revealed coarse, dark brown granules in centrilobular hepatocytes via hematoxylin and eosin (H&E) staining (Figure 2A). No signs of hemosiderin or ferritin were noted with Prussian blue staining (Figure 2B). No signs of fibrosis were observed in the liver parenchyma with Masson trichrome staining. Therefore, she was discharged with hepatotonics, and routine check-ups, including blood chemistry, are being continued. To date, a similar range of bilirubin levels is observed.

We then performed a genotype study via blood sampling. In the *ABCC2* mutation study, a missense mutation in exon 18 was discovered: arginine in gene position 768 was substituted with tryptophan (R768W). This altered the protein risk to “probably damaging” in the polyphen-2 prediction score and to “deleterious” in the SIFT (Sorting Intolerant from Tolerant) prediction model (Figure 3A). Compared to healthy individuals, our patient’s immunohistochemical staining revealed almost no multidrug resistance-associated protein 2 (MRP2) (Figure 3B).

## 3. Discussion

DJS is an autosomal recessive disorder caused by MRP2 protein dysfunction, affecting all races, nationalities, and sexes equally, although it manifests earlier in men [4,5,6,7,8]. It was first described in 1954, and it is a condition in which conjugated bilirubin is not excreted normally from hepatocytes into the bile canaliculi, resulting in hyperbilirubinemia without severe complications or inconveniences [1]. Diagnosing this syndrome is important to differentiate it from other diseases that may require further treatment. However, because of its asymptomatic nature, it is rarely detected at a young age [2]. Only in rare cases does DJS manifest as a progressive disease such as cholestasis, but even in this case, there is no risk of fibrosis or cirrhosis; thus, it rarely requires treatment [4]. As for diagnosis, in addition to hyperbilirubinemia and the characteristic gross black liver appearance, DJS patients’ urine coproporphyrin consists mostly of coproporphyrin I, whereas coproporphyrin III is most common in normal urine. Furthermore, although a histological test is not required, histological studies display dark, lysosomal, melanin-like pigment deposits, and a genotyping test will reveal *ABCC2* gene mutation [2].

There have been few reports regarding this case. In a study by Wu et al., seven patients diagnosed with DJS were studied. Genomic DNA was extracted from the blood to detect *ABCC2* mutations, and Schmorl’s staining method and immunohistochemical analysis of MRP2 expression were performed. They performed an analysis of exons 9, 10, and 14–19 and, like our study, PolyPhen-2 and SIFT were used to predict the effects of novel mutations on the protein’s function. R393W, Y1275X, p.V417I, p.G693R, p.G808V, and p.E647X were identified [9]. In a study by Toh et al., three patients diagnosed with DJS were studied. DNA from blood samples and liver tissue was isolated, and a missense mutation in gene location 2302 (C → T), accompanied by amino acid substitution R768W, was observed, as in our case [10]. In a study by Zhao et al., two patients diagnosed with DJS were studied. Whole-exome sequencing was conducted, and three variants (c.2980delA, c.1834C > T, and c.4465_4473delinsGGCCCACAG) were identified [11]. In a case report by Kamal et al., molecular testing for the *ABCC2* gene using quantitative polymerase chain reaction was used for diagnosis, and c.2273G > T, pG785V in exon 18 was identified [12]. In a case report by Gupta et al., a coproporphyrin study was performed, but the ratio of isomers was unavailable so a liver biopsy was conducted. In this report, the genotyping study was not available. [13]. 

Here, we are reporting a case where DJS was first suspected via the gross appearance of a black liver, which was initially examined via laparoscopic surgery. Subsequently, MRP2 immunohistochemical staining and genotyping were performed on a biopsy sample. Negative MRP2 immunohistochemical staining and the missense mutation R768W were found, finally diagnosing the patient with DJS. To our knowledge, this is the first case where DJS was suspected and diagnosed via the gross appearance of a black liver, phenotyping (via immunohistochemical and H&E staining), and genotyping (via detection of missense mutation). The missense mutation R768W in *ABCC2* exon 18 were mentioned in a few case studies concerning DJS patients, including our patient [14]. However, no significant treatment has been established for DJS. Therefore, hepatotonics to minimize liver injury and regular follow-ups were planned for this patient [15].

## Figures and Tables

**Figure 1 ijms-23-16168-f001:**
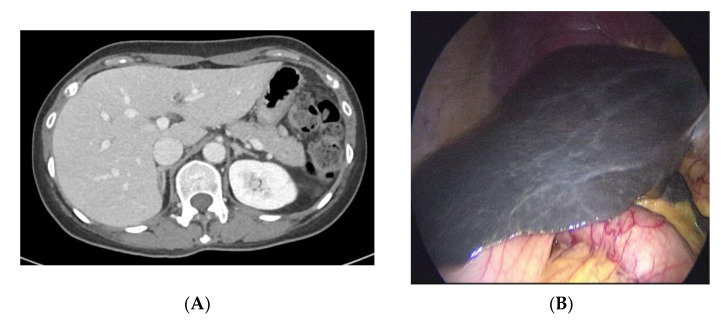
(**A**) Computed tomography findings of liver parenchyma at the time of diagnosis (axial view). (**B**) Gross findings of the patient liver observed during laparoscopic cholecystectomy.

**Figure 2 ijms-23-16168-f002:**
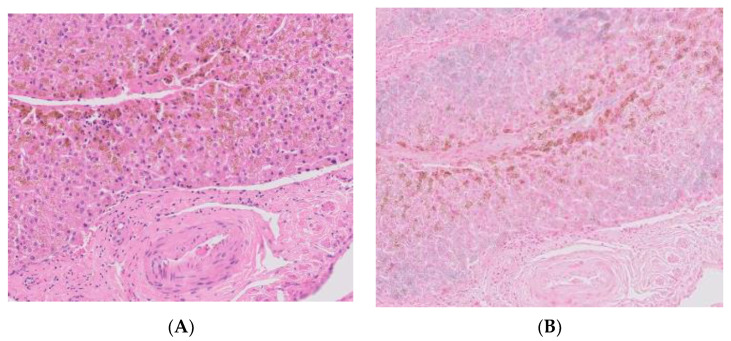
Liver biopsy specimen. (**A**) Dark brown granules in centrilobular hepatocytes are noted via hematoxylin and eosin staining. (**B**) No signs of hemosiderin deposition were noted with Prussian blue staining.

**Figure 3 ijms-23-16168-f003:**
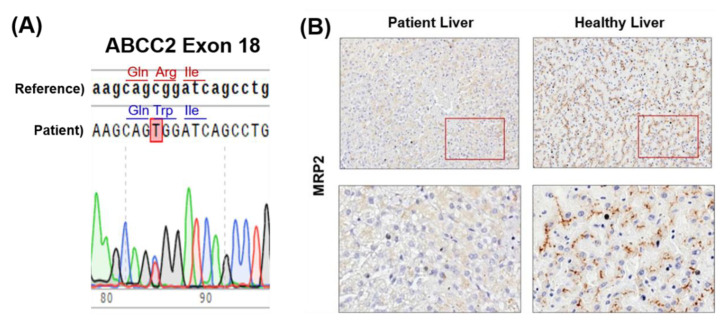
(**A**) *ABCC2* mutation study of exon 18. (**B**) Immunohistochemical staining of multidrug resistance-associated protein 2 (case patient vs. healthy subject).

## Data Availability

Data are contained within the article.

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
