# Peer review of "Genotype–Phenotype Association in ABCC2 Exon 18 Missense Mutation Leading to Dubin–Johnson Syndrome: A Case Report"

_ijms, 2022, doi:10.3390/ijms232416168_

Round 1

Reviewer 1 Report

In the present study, Kim et al. reported a case of a patient with Dubin-Johnson syndrome confirmed by a genetic study. The case is well described and instructive to readers. I have a one major concern in interpreting the result of the genotype study.

In Fig. 3A, there seems to be both ABCC2-mutant allele and wild-type allele. Is the mutation in this patient observed only in one of the two alleles? Then, why did immunohistochemistry showed almost null MRP2 staining? As DJS is an autosomal recessive disorder, the patient may have another mutation in ABCC2 gene.  

Author Response

In the present study, Kim et al. reported a case of a patient with Dubin-Johnson syndrome confirmed by a genetic study. The case is well described and instructive to readers. I have a one major concern in interpreting the result of the genotype study. In Fig. 3A, there seems to be both ABCC2-mutant allele and wild-type allele. Is the mutation in this patient observed only in one of the two alleles? Then, why did immunohistochemistry showed almost null MRP2 staining? As DJS is an autosomal recessive disorder, the patient may have another mutation in ABCC2 gene.  

  • Thank you for your comment. I am sorry for the misunderstanding. The wild type allele ii the original figure was to be seen as a reference to compare to our patient’s mutant allele. The sequencing result for our patient showed that the missense mutation in exon 18 (R768W) is homozygous. We apologize for the misunderstanding and inserted additional information to our figure to eliminate the possibility of misunderstanding (revised fig. 3A).

Reviewer 2 Report

Authors present a cases study of a patients accidentally diagnosed with Dubin-Johnson syndrome following detection of “black liver” during elective surgery for cholecystitis. Diagnosis was based on a combination of immunohistochemistry staining, genotyping, and blood test for hyperbilirubinemia. Few suggestions are below.

1.      Lines 43-44: “…which consists of the accumulation of dark and coarsely granular pigment in the centrilobular hepatocytes of the liver…”.

2.      Line 142: Please rephrase to clarify e.g., "...urine coproporphyrin consists mostly of coproporphyrin I; whereas in normal urine, coproporphyrin III..."

Author Response

Authors present a cases study of a patients accidentally diagnosed with Dubin-Johnson syndrome following detection of “black liver” during elective surgery for cholecystitis. Diagnosis was based on a combination of immunohistochemistry staining, genotyping, and blood test for hyperbilirubinemia. Few suggestions are below.

  1. Lines 43-44: “…which consists of the accumulation of dark and coarsely granular pigment in the centrilobular hepatocytes of the liver…”.
  2. Line 142: Please rephrase to clarify e.g., "...urine coproporphyrin consists mostly of coproporphyrin I; whereas in normal urine, coproporphyrin III..."

A) Thank you for your comment. We applied your recommendation to our revised manuscript as we agree to your suggestions. The phrase was changed to: ……whereas in normal urine, coproporphyrin III is the most common.